# Nanoparticle-Mediated Dual Targeting: An Approach for Enhanced Baicalin Delivery to the Liver

**DOI:** 10.3390/pharmaceutics12020107

**Published:** 2020-01-29

**Authors:** Iman Saad Ahmed, Hassan Medhat Rashed, Hend Fayez, Faten Farouk, Rehab Nabil Shamma

**Affiliations:** 1Department of Pharmaceutics & Pharmaceutical Technology, College of Pharmacy, University of Sharjah, Sharjah 27272, UAE; 2Department of Labeled Compounds, Hot Labs. Center, Egyptian Atomic Energy Authority, Cairo 11865, Egypt; hassan_vodafone@hotmail.com (H.M.R.); ango_352@yahoo.com (H.F.); 3Department of Pharmaceutics, Faculty of Pharmacy, Sinai University, Kantara 16020, Egypt; 4Department of Pharmaceutical Chemistry, Faculty of Pharmacy, Ahram Canadian University, 6th of October City, Cairo 11865, Egypt; f.farouk04@gmail.com; 5Department of Pharmaceutics and Industrial Pharmacy, Faculty of Pharmacy, Cairo University, Cairo 11561, Egypt; rehab.shamma@pharma.cu.edu.eg

**Keywords:** baicalin, lactobionic acid, chitosan lactate, nanoparticles, ionic gelation method, radiolabeling, in vivo biodistribution study, liver targeting

## Abstract

In this study, water-soluble chitosan lactate (CL) was reacted with lactobionic acid (LA), a disaccharide with remarkable affinity to hepatic asialoglycoprotein (ASGP) receptors, to form dual liver-targeting LA-modified-CL polymer for site-specific drug delivery to the liver. The synthesized polymer was used to encapsulate baicalin (BA), a promising bioactive flavonoid with pH-dependent solubility, into ultrahigh drug-loaded nanoparticles (NPs) via the ionic gelation method. The successful chemical conjugation of LA with CL was tested and the formulated drug-loaded LA-modified-CL-NPs were assessed in terms of particle size (PS), encapsulation efficiency (EE) and zeta potential (ZP) using full factorial design. The in vivo biodistribution and pharmacokinetics of the designed NPs were assessed using ^99m^Tc-radiolabeled BA following oral administration to mice and results were compared to ^99m^Tc-BA-loaded-LA-free-NPs and ^99m^Tc-BA solution as controls. Results showed that the chemical modification of CL with LA was successfully achieved and the method of preparation of the optimized NPs was very efficient in encapsulating BA into nearly spherical particles with an extremely high EE exceeding 90%. The optimized BA-loaded-LA-modified-CL-NPs showed an average PS of 490 nm, EE of 93.7% and ZP of 48.1 mV. Oral administration of ^99m^Tc-BA-loaded-LA-modified-CL-NPs showed a remarkable increase in BA delivery to the liver over ^99m^Tc-BA-loaded-LA-free-CL-NPs and ^99m^Tc-BA oral solution. The mean area under the curve (AUC_0–24_) estimates from liver data were determined to be 11-fold and 26-fold higher from ^99m^Tc-BA-loaded-LA-modified-CL-NPs relative to ^99m^Tc-BA-loaded-LA-free-CL-NPs and ^99m^Tc-BA solution respectively. In conclusion, the outcome of this study highlights the great potential of using LA-modified-CL-NPs for the ultrahigh encapsulation of therapeutic molecules with pH-dependent/poor water-solubility and for targeting the liver.

## 1. Introduction

Baicalin (BA)—a well-known herbal medicine officially listed in the Chinese Pharmacopeia—is traditionally used to treat a wide range of diseases such as psoriasis, atopic dermatitis, asthma, bronchitis, hepatitis, nephritis, hypertension, inflammation and others [1,2]. The use of BA in the treatment of these different conditions is due to its reported antibacterial [3], antifungal [4], antiviral [5], anti-inflammatory [6], antipyretic [7], antihypertensive [8], sedative [9] and antithrombotic activity [10]. Recently, BA was also reported to exert antioxidant [11], hepatoprotective [12], anticancer [13] and neuroprotective [14] action.

Because of the overwhelming medical benefits of BA and its potential to be used as a novel drug in the market, it is important to address some its pharmaceutical limitations such as its low water solubility and poor pharmacokinetic properties that may hamper its translation to actual market. For instance, BA is classified as class II drug according to the Biopharmaceutical Classification System [15] and exhibits very poor oral bioavailability (2.2%), extremely short biological half-life (0.75 h) and complex metabolic pathway [16,17].

Several studies attempted to improve the oral bioavailability of BA through formulation using different delivery platforms such as oral chewable tablets [18,19], microemulsions [20], self-assembled nanoparticles [21], nanocrystals [22], solid dispersions [15,23], solid lipid nanoparticles [24,25,26] and thermosensitive hydrogels [27].

Designing novel materials for biomedical applications generally requires the use of biodegradable ones. Chitosan, a renewable natural polysaccharide, is of great interest due to its good biocompatibility and biodegradability [28,29]. Chitosan is also used for hepatocytes attachment due to structure similarity to glycosaminoglycans, which are components of the liver extracellular matrix [30,31].

Hepatocellular accumulation of nanoparticles (NPs) can be enhanced via receptor mediated endocytosis [32,33]. In this respect, several ligands such as RGD peptide, transferrin, phenylboronic acid, folic acid and lactobionic acid have been reported to enhance the target cell recognition and uptake [34,35,36,37].

Asialoglycoprotein receptor (ASGPR) is a hepatic C-type lectin that is responsible for carbohydrate recognition [38], specifically, the galactose or N-Acetylglucosamine residues. Under normal conditions ASGPR is expressed on the sinusoidal and basolateral surface of the plasma hepatocyte membrane [39]. This receptor is over expressed in case of liver diseases such as cancer and cirrhosis [40]. ASGPR receptors are used as a liver tissue portal by viruses such as hepatitis A and hepatitis B viruses [38] and are targeted by radiopharmaceuticals for liver function imaging [41].

Since ASGPR binds normally to galactose, lactobionic acid (which is composed of galactose and gluconic acid) is a good candidate to enhance drug delivery to the liver [42]. In this context, LA-modified dendrimer entrapping gold NPs were specifically uptaken by hepatoma carcinoma cell lines [43]. Similarly, polyethylene glycol-linked LA (PEG-LA) was conjugated onto the surface of laponite nanodisks for the targeted delivery of doxorubicin (DOX) to liver cancer cells [44]. These results proved that LA-modified carriers display a significantly higher therapeutic efficacy in inhibiting the growth of hepatocellular carcinoma cells (HepG2 cells) than non-targeted ones [45].

Several types of research have used radiolabelling as a tracking technique for investigating the fate and pharmacokinetics of novel drug delivery systems in the body [46,47,48]. Advantages of such technique include the ability to follow the dispersion of a formulation when administered intraocular, intranasal, orally or through other routes in almost any physical form [49,50,51].

The aim of this work is to develop and characterize ultrahigh BA-loaded-LA-modified-CL-NPs that can increase the site-specific delivery of incorporated BA to the liver. First, LA, as a liver-targeting moiety, was reacted with the amino groups of chitosan lactate (CL) using ethyl dimethylaminopropyl carbodiimide (EDC) to form dual liver-targeting LA-modified-CL polymer. Then, BA was encapsulated into LA-free-CL-NPs or LA-modified-CL-NPs using the ionic gelation method to test the targeting ability of the designed NPs. The effect of several formulation and process variables on the particle size (PS), zeta potential (ZP), and encapsulation efficiency (EE) of the prepared NPs were investigated using full factorial design. Finally, in vivo biodistribution studies were carried out using ^99m^Tc-radiolabeled-BA to assess the pharmacokinetics and the liver-targeting ability of ^99m^Tc-BA-loaded-LA-modified-CL-NPs compared to ^99m^Tc-BA-loaded-LA-free-CL-NPs and ^99m^Tc-BA solution as a control following oral administration of equal doses to mice.

## 2. Materials and Methods

### 2.1. Materials

Chitosan oligosaccharidelactate (CL, M_n_ = 5000, deacetylation degree > 90%), Baicalin (BA; (2S,3S,4S,5R,6S)-6-(5,6-dihydroxy-4-oxo-2-phenyl-chromen-7-yl)oxy-3,4,5-trihydroxy-tetrahydropyran-2-carboxylic acid), lactobionic acid (LA; 4-*O*-β-galactopyranosyl-D-gluconic acid), 1-ethyl-3-(3-dimethylaminopropyl)carbodiimide hydrochloride (EDC), sodium tripolyphosphate (TPP) and cellulose dialysis bag with molecular weight cut-off (MWCO) of 14 kD were purchased from Sigma Aldrich (St Louis, MN, USA). All other chemicals were of analytical grade. Technetium-99m was eluted as ^99m^TcO_4_^−^ from ^99^Mo/^99m^Tc generator, Radioisotope Production Facility, Cairo, Egypt.

### 2.2. Preparation of BA-loaded-LA-Modified-CL-NPs

To explore the possibility of developing dual-targeted polymeric therapeutic NPs intended to enhance the localization of BA into the liver and reduce its systemic delivery, LA was coupled with CL to form LA-modified-CL. LA-modified-CL was then used in the formation of ultrahigh drug-loaded NPs to target the liver. LA was coupled with CL through the interaction of the carboxyl groups of LA with the amino groups of CL using EDC [52,53,54]. BA was also encapsulated into LA-free-CL-NPs as a control. BA-loaded-LA-modified-CL-NPs and LA-free-BA-loaded-CL-NPs were prepared using the ionic gelation method reported by Calvo et al. [55].

To form LA-free-BA-loaded-CL-NPs, 10 mg of BA was dissolved in 2.0 mL aqueous solution containing 0.25% (*w/v*) TPP (pH = 9.8) and the resulting solution was added under gentle magnetic stirring (750 rpm; Stuart SB161, UK) to 10 mL aqueous solution containing 1% (*w/v*) CL (pH = 4.7).

To form BA-loaded-LA-modified-CL-NPs, EDC (100 µL; 10 mg/mL) and different amounts of LA were added to 1% CL aqueous solution (pH 4.7) and the coupling reaction was left to react for 1 h. This allows the activation of the carboxylic groups of LA and the formation of the O-acylisourea intermediate which then rapidly reacts with the primary amino groups that are available on CL to form the new amide bond [56,57].

Coupling was investigated either before the addition of BA-TPP solution (i.e., before NPs formation) or after the addition of BA-TPP solution (i.e., after NPs formation). In either way, the volume ratio of CL: BA-TPP solution was kept constant at 5:1, yielding 12 mL total volume per run. The mixture was further stirred for 30 min at room temperature to allow for the spontaneous formation of the NPs. The best coupling condition and the amount of LA to be used were selected statistically using a factorial design.

In order to perform different characterization tests, the formed NPs were separated from their aqueous dispersions by centrifugation at 20,000 rpm at room temperature for 30 min using a high-speed centrifuge (Hermle Labortechnik GmbH, Germany).

### 2.3. Optimization of BA-Loaded-LA-CL-NPs

A 2^1^ × 3^1^ full factorial experimental design was employed to investigate the influence of the process and formulation variables on the characteristics of BA-loaded-LA-modified-CL-NPs using Design-Expert-7 Software. Two numeric factors have been set for the experimental design: 1) the method of addition of LA-EDC to CL solution at two levels (X1: before the addition of BA-TPP or after the addition of BA-TPP) and 2) the resulting concentration of LA, obtained upon adding different amounts of LA to CL solution, at three levels (X_2_: 50 mg/L, 75 mg/L or 100 mg/mL). Based on that, central composite design was found to be suitable for planning the proposed formulations in the form of six runs (6 formulations). On the other hand, three responses were adopted to be tracked for the optimization of the studied factors: 1) percentage entrapment efficiency (Y_1_: EE%), 2) particle size (Y_2_: PS) and 3) zeta potential (Y_3_: ZP). The characteristics of the prepared formulations (F1→6) are shown in Table 1. All six formulations were performed in triplicate randomized way to satisfy the statistical requirements. The polydispersity index (PDI) of all formulations was also measured.

### 2.4. Selection of the Optimized Formulation

The overall desirability value was chosen as the differentiating parameter to compare the six formulations [58]. Furthermore, the values of the dependent variables were optimized with optimization criterion set at the highest EE%, the smallest PS and the highest ZP to yield the system with the highest overall desirability factor. Following selection, the optimized formulation was subjected to further modification and characterization.

### 2.5. Characterization of BA-loaded-LA-Modified-CL-NPs (F1→6)

#### 2.5.1. Particle size and Polydispersity

The particle size (PS) and size distribution of freshly prepared BA-loaded-LA-modified-CL-NPs and BA-loaded-LA-free-CL-NPs were determined by photon correlation spectroscopy dynamic light scattering using Zetasizer 2000 Malvern Instruments, U.K. (Nano-ZS; Malvern Instruments, Malvern, UK). All measurements were performed using a clear disposable zeta cell at approximately 25 °C and an angle of the laser incidence of 173°. Mean values for each preparation were obtained by at least duplicate measurements of three different batches. Results were given as the mean particle diameter (z-average) and polydispersity index (PDI).

#### 2.5.2. Determination of Zeta Potential

The zeta potential (ZP) was measured using a ZetaSizer Nano ZS (Malvern Instruments, Worcestershire, UK), making use of Laser Doppler Velocimetry. A suitable amount of the freshly prepared NPs was placed into the electrophoretic cell of the instrument at approximately 25 °C and laser light scattered at an angle of 17° was used. Mean values for each preparation were also obtained by at least duplicate measurements of three different batches.

#### 2.5.3. Encapsulation Efficiency (EE%)

Following preparation of BA-loaded NPs, the NPs were separated from the aqueous medium by centrifugation at 20,000 rpm at room temperature for 30 min. The concentration of entrapped BA was measured indirectly by measuring the concentration of BA in the supernatant spectrophotometrically at λ 275 nm (UV-1800 PC, Shimadzu, Kyoto, Japan). The BA encapsulation efficiency (EE%) was calculated using the following equation:(1)$$EE\%=\frac{Total\;BA-Free\;BA}{Total\;BA}\times 100$$

### 2.6. Physico-Chemical Characterization of Optimized BA-loaded-LA-Modified-CL-NPs

Based on the optimization results, formulation 4 (Table 1) was selected for further modification, physico-pharmaceutical evaluation, stability testing and in vivo studies.

#### 2.6.1. Attenuated Total Reflectance-Infrared (ATR-IR) Spectroscopy

The physico-chemical interactions between the drug and excipients, the interaction between LA and CL in addition to the entrapment of BA in NPs were studied by ATR-IR spectroscopy using ATR-IR spectrophotometer (IRspirit, Schimadzu, Japan). ATR-IR scanning was performed for BA, CL, LA, BA-loaded-LA-free-CL-NPs and BA-loaded-LA-modified-CL-NPs. 2-3 mg of each sample was mixed with 100 mg of potassium bromide and compressed into thin discs using a hydrostatic press. Finally, the prepared samples were scanned with ATR-IR spectrophotometer over a wavelength range from 4000–400 cm^−1^.

#### 2.6.2. In Vitro Release Studies

The in vitro release of BA from its aqueous solution in 0.25% TPP, BA-loaded-LA-free-CL-NPs and optimized BA-loaded-LA-modified-CL-NPs was determined using the dialysis bag method [59]. To study the solubilizing effect of TPP on BA, the release of BA from its suspension in distilled water was also determined. Cellulose dialysis bags (6 cm × 2.2 cm with MWCO of 14 kD, Sigma Chemical Company, USA) representing the donor compartment were used to retain the tested samples while allowing the soluble BA to permeate into the release medium. Initially, a volume equivalent to 0.84 mg BA was taken from each of the four tested dispersions. These volumes were then pipetted into the dialysis bags. Sealed dialysis bags were then placed into 100 mL phosphate buffer saline (PBS; pH 6.8) representing the receptor compartment. The temperature was maintained at 37 °C in an incubator shaker (Stuart SBS 40, Staffordshire, UK) with a stirring rate of 100 stroke per min. Samples of 3 mL were collected from the receptor media at 0.5, 1, 2, 3, 4, 6, 8 and 12 h and were immediately replaced by equal volume of fresh medium. The samples were then filtered and analyzed for drug content spectrophotometrically at λ 275 nm. A standard curve of BA in PBS (pH 6.8) was generated over the range of 0.1–50 µg/mL and used to convert absorbance to concentration. A cumulative release profile was generated by normalizing the data against the total amount of BA and reported as percentage drug release. All release experiments were conducted in triplicates (*n* = 3).

#### 2.6.3. Transmission Electron Microscopy

The morphology of the optimized BA-loaded-LA-modified-CL-NPs and BA-loaded-LA-free-CL-NPs were examined using transmission electron microscopy (JEOL JEM1230, Tokyo, Japan). One-drop of the NPs suspension was placed on a carbon coated film 300 mesh copper grid, and afterwards a drop of ethanol was added to solidify their morphology before microscopic observation.

### 2.7. Short-Term Stability Study

The short-term stability of the developed BA-loaded-LA-modified-CL-NPs and BA-loaded-LA-free-CL-NPs was evaluated and was carried out by storing the NPs at room temperature or at 4 °C in refrigerator for 15 days. The vials containing BA-loaded-NPs suspensions were sealed and wrapped in aluminum foil and subdivided into two groups. One group was stored at room temperature (25 °C) and the other group was stored in refrigerator at 4 °C for 15 days. Following 15-day storage, aliquots were taken from the vials and subjected to PS, EE% and ZP analysis as described above. The change in appearance (presence of aggregates), PS, EE% and ZP were recorded and compared to results obtained from freshly prepared NPs.

### 2.8. In Vivo Studies

In vivo biodistribution studies were carried out in mice using a radiolabeling technique to study the ability of the developed BA-loaded-LA-modified-CL-NPs to target the liver in comparison to BA-loaded-LA-free-CL-NPs, and BA solution in TPP as a control following oral administration of equal doses.

#### 2.8.1. Preparation of ^99m^Tc-BA complex

Direct radiolabelling method was applied to radiolabel BA with ^99m^Tc under reductive conditions using stannous chloride (SnCl_2_) as a reducing agent [60]. The individual effect of different labelling conditions on the efficiency of the radiolabelling technique were first explored and optimized to maximize the percentage radiochemical yield (% RCY) of ^99m^Tc-BA complex. The reaction variables studied were: BA quantity (0.3-3 mg), SnCl_2_ quantity (0.1-2 mg), pH of the reaction solution (3-11) and the reaction time (5-90 min). Ascending paper chromatography was performed to assess the efficiency of the radiolabelling process using dual-solvent system as mobile phase and strips of Whatman paper No.1. Free ^99m^TcO_4_^−^ was determined using acetone as a mobile phase, while 0.5 M NaOH was the mobile phase of choice to determine ^99m^Tc-colloidal impurities [61,62]. Labelling optimization experiments were performed as follows: different amounts of BA were first dissolved in 1 mL DMSO and then mixed with 1 mL ^99m^TcO_4_^−^ (72 MBq) solution freshly eluted from ^99^Mo/^99m^Tc generator (100 μL of ^99^Mo/^99m^TC generator eluate contains 7.2 MBq of ^99m^TcO_4_^−^). This solution was then added to an evacuated vial containing different amounts of SnCl_2_ powder. Finally, 1 mL of absolute ethanol was added and the reaction mixture was shaken well and left to react for different times at room temperature. The pH of the reaction medium was adjusted over the studied pH range by adding 0.1 N HCl or 0.1 N NaOH solutions. All experiments were conducted in triplicates for each studied factor and the mean values ± SD were calculated. Statistical multiple comparisons were made using one-way analysis of variance (ANOVA) F-test and a *p*-value of ≤ 0.05 was considered statistically significant.

The optimized radiolabeled ^99m^Tc-BA complex was then used to produce optimized ^99m^Tc-BA-loaded-LA-modified-CL-NPs and ^99m^Tc-BA-loaded-LA-free-CL-NPs using the same procedure described before. In addition, assessment of the in vitro stability of the resulting complex for 24 h was performed as described by Motaleb et al. [63]. The EE%, PS, ZP and in vitro drug release of the radiolabelled NPs were evaluated to study the effect of BA radiolabelling on the NPs characteristics before using them in in vivo studies.

#### 2.8.2. Animals

Fourty-five male Swiss albino mice, weighing 20-30 g, were used throughout the study. The animals were brought from the animal house of the Faculty of Pharmacy, Cairo University, Egypt. The animals were housed in plastic cages at the Animal Care Facility under controlled conditions of temperature (25 °C) and humidity (65%) with a 12 h on/off light cycle. The animals were fed with standard mice diet with free access to water throughout the entire period of the study. The study was approved by the Institutional Animal Ethics Committee at Cairo University (REC-PT-1495) on 13-06-2019 and the animals were treated according to the principles of laboratory animal care and use. The Egyptian Atomic Energy Authority guidelines for animal experiments were followed. All animal experiments also comply with Directive 2010/63/EU.2.7.1.

#### 2.8.3. Study Design and Drug Administration

In vivo studies were performed using a non-blind, three-treatment, randomized, parallel design. On the study day, the animals were randomly divided into three treatment groups of fifteen mice each (*n* = 15). The three treatments were as follows: Group A received ^99m^Tc-BA solution (control group), Group B received ^99m^Tc-BA-loaded-LA-modified-CL-NPs and Group C received ^99m^Tc-BA-loaded-LA-free-CL-NPs. A volume equivalent to the dose was administered to the conscious animals through the oral route using an intragastric tube. The dose of BA given in each treatment was 0.007 mg BA/g body weight.

#### 2.8.4. Pharmacokinetic and Biodistribution Studies

Following drug administration, three mice from each group were anesthetized by chloroform and sacrificed at each of the following time points: 1, 2, 4, 6, and 24 h. The organs/tissues were dissected, washed with normal saline, cleaned from any adhering materials and finally weighed. Blood samples were also collected and weighed. The radioactivity uptake by each organ as well as the background was measured using a gamma counter (ST360 Radiation Counter, Spectrum Techniques, USA). The data was recorded as mean percentage administered dose per gram (%AD/g ± SD) of three mice for each time interval. The mean BA radioactivity uptake (%AD/g) in blood and liver was plotted against time (h). The total amount of blood radioactivity was calculated given that blood constitutes 7% of the mouse weight [64,65]. Pharmacokinetic characteristics of BA from blood and liver data were estimated for each mouse by using WinNonlin software program (version 1.5, Scientific Consulting Inc., Cary, NC, USA). Non-compartmental analysis was adopted. The maximum radioactivity uptake of the administered dose per gram tissue was taken as the C_max_ and the total radioactivity uptake of the administered dose per gram tissue over the 24 h period was used to estimate the total area under the curve from 0 to 24 h (AUC _0-24_, %AD/g). Differences between pairs of means were performed on C_max_ and AUC _0-24_.

The ability of the administered NPs to deliver BA specifically to the liver following oral administration was estimated by calculating the drug targeting index (DTI), the drug targeting efficiency (DTE) and the relative targeting efficiency using the following equations [66]:(2)$$DTI=\frac{Liver\;concentration\;of\;BA-loaded-LA-modified-CL-NPs\;at\;time\;\left(t\right)}{Liver\;concentration\;of\;LA-free\;BA-loaded-CL-NPs\;at\;time\;\left(t\right)}$$
(3)$$DTE=\frac{Liver\;AUC_{0-24}}{Blood\;AUC_{0-24}}$$
(4)$$RTE=\frac{Liver\;AUC_{0-24\;}of\;BA-loaded-LA-modified-CL-NPs}{Liver\;AUC_{0-24\;}of\;LA-free\;BA-loaded-CL-NPs}$$

### 2.9. Statistical Analysis

All in vitro measurements were carried out in independent triplicates and values are presented as mean ± SD unless otherwise noted. Statistics were carried out using Minitab 16 (UK). For comparisons between two groups, two-tailed unpaired Student’s t-test was employed. For multiple comparisons, one-way analysis of variance (ANOVA) followed by Tukey’s post-hoc test was utilized. A *p*-value of ≤ 0.05 was considered statistically significant.

## 3. Results and Discussion

### 3.1. Optimization of BA-loaded-LA-Modified-CL-NPs

BA-loaded-LA-modified-CL-NPs with extremely high EE% were prepared successfully using the above-mentioned optimized technique. To our knowledge, this is the first study to report the successful use of CL instead of widely used chitosan to encapsulate BA in CL-NPs with a very high EE% exceeding 90%. CL in this case was very useful because BA would precipitate immediately in acidic media that must be otherwise used to solubilize chitosan during NPs formation. On the contrary, CL aqueous solution has a pH of 4.7 thus preventing the precipitation of BA and helping the efficient entrapment of BA inside the NPs.

Factorial designs are commonly used to analyze the influence of different formulation and process variables on the characteristics of the developed drug delivery system. The effect of the method of addition of LA-EDC to CL solution and the concentration of LA used on the PS, EE% and ZP of the prepared NPs were statistically analyzed using 2^1^ × 3^1^ full factorial experimental design. Table 1 summarizes the characteristics of the six BA-loaded-LA-modified-CL-NPs formulations prepared using this design. The predicted R^2^ values were in a reasonable agreement with the adjusted R^2^ of all responses (Table 2).

Adequate precision measures the signal-to-noise ratio to ensure that the model can be used to navigate the design space. A ratio greater than 4 (the desirable value) was observed in all responses. Graphical analysis of the effects of the variables on EE%, PS and ZP are shown in Figure 1.

The percentage BA entrapped within the BA-loaded-LA-modified-CL-NPs varied from 80% to 93% (Table 1). Statistical analysis showed that only the method of addition of LA-EDC to CL solution had a significant impact on the drug EE% (*p*-value = 0.0001). As depicted in Figure 1A addition of LA-EDC to CL before adding BA-TPP solution (i.e., before NPs formation) resulted in a significant increase in the percentage drug entrapped within the NPs. This could be attributed to reduced available amino groups of CL due to conjugation with LA, which may result in slower formation of NPs due to slower interaction between amino groups of CL and phosphate groups of TPP thus allowing efficient entrapment of BA from TPP solution. The slow interaction between TPP and CL might as well help in keeping BA in a soluble form as our previous results show that phosphate groups of TPP play an essential role in increasing the water solubility of BA [27].

The average PS, presented as z-average diameter, of BA-loaded-LA-modified-CL-NPs ranged from 621 nm to 2250 nm with PDI values ranging from 0.5 to 0.9 (Table 1). As shown in Table 2, both tested variables had a significant positive impact on the mean PS (*p* ˂ 0.05). Addition of LA-EDC to CL before adding BA-TPP resulted in a significant increase in the average PS of the prepared NPs (p = 0.0004) as depicted in Figure 1B. This could be attributed to the observed increase in the EE% of the drug as discussed before. Increasing the concentration of LA simultaneously increased the PS (p < 0.0001). The positive relationship of LA concentration on the PS might be attributed to the increased consumption of the amino groups of CL by the addition of higher concentration of LA. This might further slowdown the rate of NPs formation between CL and TPP thus resulting in larger PS. Increasing the concentration of LA might as well increase the viscosity of CL aqueous solution, which provides higher resistance to the diffusion of TPP solution into the CL aqueous phase and larger NPs are thus formed. The use of higher concentration of LA might as well decrease the pH of CL solution thus leading to decreased ionization of TPP, which could negatively affect its crosslinking with CL. The decrease in the pH of CL solution might in addition increase the electrostatic repulsive forces between the polymer chains of CL due to protonation of amino groups, which could hinder the easy formation of the NPs thus resulting also in larger particles.

The ZP values of the prepared NPs were in the range of 48 to 51 mV. The high positive charge acquired by the prepared BA-loaded-LA-free-CL-NPs and BA-loaded-LA-modified-CL-NPs indicated the abundance of freely ionized amino groups of CL on the surface of the NPs which is in agreement with the polycationic nature of chitosan [67]. As presented in Table 2 and graphically illustrated in Figure 1C, the increase in the concentration of LA was associated with a significant decrease in the ZP (p = 0.0016) while the method of addition of LA had no effect. This could be attributed to the increased interaction between the carboxyl groups of LA and the amino groups of CL with the addition of higher concentration of LA, thus leading to the reduction of the total number of free ionized amino groups of CL and the ZP.

The 2-way interaction between the two independent variables had a slight effect on EE% but a more significant one on PS (*p* = 0.0343 and *p* = 0.0014 respectively) thus highlighting the complexity of formulation.

The desirability function combines all the responses into one variable to predict the optimum levels for the studied factors [68]. Accordingly, desirability was calculated to select the optimized formulation with the smallest PS along with the highest EE% and highest ZP. The highest desirability value (0.688) was depicted in the system produced by adding LA-EDC to CL solution before the addition of BA-TPP solution to result in a concentration of 50 mg/mL of LA (F4 in Table 1). It is worth noting that BA-loaded-LA-free-CL-NPs showed an average PS of 485 nm, EE% of 77.4%, ZP of 51.6 mV and PDI of 0.78, which clearly indicates that conjugation of LA to CL plays a very significant role in increasing the EE% of BA as explained before (Table 1). Conjugation of LA to CL also resulted in a significant increase in PS (p<0.05) with no effect on ZP or PDI. In an attempt to further reduce the PS, the optimum formulation (F4) was sonicated for 1 min using a bath sonicator (Crest Ultrasonics Corp., Trenton, NJ, USA). The sonicated F4 BA-loaded-LA-modified-CL-NPs was further evaluated for EE%, PS, ZP and PDI and results were compared to sonicated BA-loaded-LA-free-CL-NPs. The effect of sonication on the EE%, PS, and ZP of BA-loaded-LA-free-CL-NPs and optimized BA-loaded-LA-modified-CL-NPs are described in Table 3. Results showed that sonication lead to a significant decrease in PS and PDI of BA-loaded-LA-modified-CL-NPs (*p* = 0.001) but had no effect on EE% and ZP. Sonication, on the other hand, had a moderate effect on the PS of BA-loaded-LA-free-CL-NPs (*p* = 0.01) with no effect on the other NPs characteristics.

### 3.2. Attenuated Total Reflectance-Infrared ATR-IR Spectroscopy

The proposed interaction mechanism between LA, EDC, and CL during the formation of LA-modified-CL is presented in Figure 2.

The reaction of carboxylic acid groups of LA **(I)** with the amino groups present on the surface of CL was achieved via cross linking using the zero-length cross linker (EDC) **(II)**. The proposed scheme for the reaction is based on the ability of EDC to interact with the carboxylic acid moiety of LA **(I)** to form an adduct **(III).** The adduct then directly reacts with the primary amine of CL **(IV)** to generate an amide-based conjugate **(V)**. This method of conjugation consumes no organic solvents, which makes it an almost green synthesis method.

The successful interaction was investigated by ATR-IR spectroscopy (Figure 3). Figure 3a illustrates the ATR-IR spectrum of LA. The spectrum reveals the C-O-C symmetric and asymmetric stretching vibration bands at 1020 and 1272 cm^−1^. The bending vibration of the LA appears at 1426 and 1463 cm^−1^ for the alcoholic and carboxylic acid O-H, respectively. The carboxylic acid C=O stretching vibration appears at 1743 cm^−1^. Figure 3b represents the spectrum of CL. The spectrum reveals the C-O-C stretching appearing at 1036 and 1071 cm^−1^ for symmetric and asymmetric stretching vibrations, respectively. The alcoholic O-H stretching appears at 3364 cm^−1^ while its bending vibration appears at 1379 cm^−1^. The amide C=O of CL appears at 1650 cm^−1^. The free NH_2_ appears around 1650 (overlapping with that of C=O) and 1509 cm^−1^. In the fingerprint region, the NH_2_ displays a bending vibration at 898 cm^−1^. These NH_2_ corresponding vibrations are absent (1509 and 898 cm^−1^) in the BA-loaded-LA-modified-CL-NPs, which indicates the consumption of the NH_2_ in conjugation with LA (Figure 3c), while they are present in the BA-loaded-LA-free-CL-NPs (Figure 3d). The inclusion of the drug was tested by screening for the characteristic aromatic peaks of BA. Figure 3e reveals the C–C stretch appearing at 1475 cm^−1^ and the C-H out of plane bending appearing at 682 cm^−1^. These bands appear on both BA-loaded-LA-modified-CL-NPs and BA-loaded-LA-free-CL-NPs indicating the inclusion of the drug inside the NPs.

### 3.3. In Vitro Release Studies

The in vitro release of BA from BA-loaded-LA-free-CL-NPs and optimized BA-loaded-LA-modified-CL-NPs in comparison to BA release from its aqueous solution in 0.25% TPP and its suspension in distilled water are shown in Figure 4.

Drug release from NPs depends on the dissolution of the encapsulated drug, its diffusion out of the polymeric NPs in addition to NPs swelling and erosion/degradation in the release medium.

The release profile of BA from its suspension in distilled water showed less than 20% of the drug was released over the first 2 h with only 42% of the drug being released after 12 h. In contrast, the release of the drug was significantly enhanced in presence of TPP or when the drug was incorporated in NPs (p ≤ 0.0001); although the latter showed that the NPs played a role in slowing down the release over the first 6 h. These results are in agreement with our previous results in that phosphate groups may it be in TPP or phosphate buffer play an essential role in increasing the water solubility of BA and hence its release/dissolution [27]. Results also show that the rate and extent of drug release from BA-loaded-LA-modified-CL-NPs were lower compared to BA-loaded-LA-free-CL-NPs. For instance, the mean percentage drug release from BA-loaded-LA-free-CL-NPs after 2 h was about 67% compared to only 55% from BA-loaded-LA-modified-CL-NPs. This could be attributed to the reduced water uptake and subsequent reduced swelling of LA-modified-CL-NPs in the release buffer due to consumption of hydrophilic amino groups of CL as a result of complexation with LA. This will result in less porous NPs leading to reduced drug diffusion out of the polymer. Another explanation might arise from an increased mechanical resistance to erosion/degradation of polymeric LA-modified-CL compared to LA-free-CL thus resulting in slower drug release.

### 3.4. Morphological Characterization

TEM micrographs of BA-loaded-LA-free-CL-NPs and optimized BA-loaded-LA-modified-CL-NPs are shown in Figure 5.

TEM micrographs showed that both formulations resulted in the formation of nearly spherical NPs with distinct localized gelatinization layer of CL around the surface of each particle. TEM results confirmed the particle diameter to be as obtained by the size analyzer and were in agreement with previous results in that conjugation of LA to CL resulted in a significant increase in PS.

### 3.5. Short-term stability studies

Results from short-term stability studies showed an insignificant change in the PS, EE% and ZP for both BA-loaded-LA-free-CL-NPs and BA-loaded-LA-modified-CL-NPs when stored in refrigerator at 4 °C for 15 days (Table 4). A significant increase in the PS of both types of NPs was however observed after 15-day storage at room temperature (Figure 6) accompanied with a significant decrease in the ZP but no change in EE%. To eliminate possible changes in the PS upon standing all batches were freshly prepared and used either on the same day or stored for no more than 15 days at 4 °C.

### 3.6. In vivo Pharmacokinetic and Biodistribution Studies

#### 3.6.1. Radiolabelling of BA

The effect of different labelling conditions on the percentage radiochemical yield (% RCY) of ^99m^Tc-BA complex is depicted in Figure 7. The maximum % RCY of ^99m^Tc-BA complex obtained following the optimization process was 89.8 ± 1.1%. This optimum yield was obtained when 2 mg of BA and 0.5 mg SnCl_2_ were used and the radiolabeling reaction was left for 45 min at pH 7 at room temperature (25 °C). ^99m^Tc-BA complex was stable in vitro for at least 24 h. None of the physico-chemical characteristics of the NPs such as PS, EE%, ZP, PDI and in vitro release were affected by using radiolabeled BA in the formation of the NPs.

#### 3.6.2. Liver-Targeting Assessment

The liver-targeting potential of ^99m^Tc-BA-loaded-LA-modified-CL-NPs relative to ^99m^Tc-BA-loaded-LA-free-CL-NPs and ^99m^Tc-BA solution following oral administration of equal doses to mice was assessed by performing biodistribution studies. The biodistribution of ^99m^Tc-BA from the three treatments was studied by determining the radioactivity uptake in the different tissues at different time intervals up to 24 h. The biodistribution results in liver tissue and blood are described in Figure 8 and Figure 9, while the pharmacokinetic characteristics (C_max_ and AUC_0–24_) and hepatic specific delivery parameters are displayed in Table 5 and Table 6.

Remarkable differences in the biodistribution of ^99m^Tc-BA in the different tissues from the three treatments were observed. Biodistribution results in liver tissue showed significantly higher ^99m^Tc-BA concentration from ^99m^Tc-BA-loaded-LA-modified-CL-NPs at different time intervals compared to ^99m^Tc-BA-LA-free-loaded-CL-NPs and ^99m^Tc-BA solution.

Liver specific delivery parameters obtained from BA-loaded-LA-modified-CL-NPs were also superior compared to BA-loaded-LA-free-CL-NPs. For instance, at 2 h following oral administration, the drug targeting index (DTI) was found to be nearly four-times higher from LA-modified-BA-loaded-CL-NPs compared to BA-loaded-LA-free-CL-NPs indicating the ability of LA-modified-CL-NPs to target BA to the liver. The average drug targeting efficiency (DTE) was estimated to be 4.22 from BA-loaded-LA-modified-CL-NPs, indicating that the amount of BA in the liver was more than four-times higher than that found in the blood. However, the average DTE from BA-loaded-LA-free-CL-NPs was about 1.44 indicating there was almost 1.5-time more drug in the liver than the blood.

It is not surprising that BA-loaded-LA-free-CL-NPs were able to deliver more BA to the liver compared to blood, which is consistent with previous results obtained by some of the authors of this work. Previously, it was shown that encapsulation of atorvastatin calcium (AC) in NPs ≥ 200 nm improved induced hyperlipidemia in rats with no adverse effects despite their lower bioavailability compared to Lipitor. These findings were attributed to increased liver uptake of larger AC-NPs resulting in apparent lower bioavailability in blood but better liver targeting unlike smaller particles AC-NPs < 100 nm which showed higher bioavailability due to preferential circulation in the plasma, mediocre liver targeting and increased toxicity ([69,70,71].

The DTE results indicate there was almost three-fold increase in the targeting efficiency of the NPs due to conjugation of LA to CL compared to LA-free NPs. The relative targeting efficiency (RTE) from BA-loaded-LA-modified-CL-NPs was also more than double (2.24) that obtained from BA-loaded-LA-free-CL-NPs. Overall, these results indicate the more efficient site-specific delivery of BA from optimized BA-loaded-LA-modified-CL-NPs to the liver compared to BA-loaded-LA-free-CL-NPs. BA-loaded-LA-free-CL-NPs can deliver BA to the liver, due to preferential liver uptake, but to a much lesser extent compared to BA-loaded-LA-modified-CL-NPs.

With regard to extent parameters in the blood obtained from BA oral solution, the mean C_max_ estimates from BA-loaded-LA-free-CL-NPs and BA-loaded-LA-modified-CL-NPs were determined to be 400% and 350% of that obtained from the oral solution respectively. The mean blood AUC_0–24_ estimate from BA oral solution, which reflects the total amount of drug absorbed over the 24 h period, was determined to be as low as 4% and 5% relative to the mean AUC_0–24_ estimates obtained from BA-loaded-LA-free-CL-NPs and BA-loaded-LA-modified-CL-NPs respectively. These figures indicate that encapsulation of BA in CL-NPs increases its relative oral bioavailability to a great extent compared to oral solution.

Regarding extent parameters in the liver, the mean C_max_ estimates from BA-loaded-LA-free-CL-NPs and BA-loaded-LA-modified-CL-NPs were determined to be 342% and 742% relative to that obtained from the oral solution respectively. While the mean liver AUC_0–24_ estimate from BA oral solution was determined to be as low as 8.5% and 3.7% relative to that obtained from BA-loaded-LA-free-CL-NPs and BA-loaded-LA-modified-CL-NPs respectively.

Based on these results, it can be concluded that encapsulation of BA in LA-modified-CL-NPs significantly improves its local bioavailability in the liver relative to blood, which is preferred due to high localization of therapeutic agent at the site of action which might result in improved efficacy and reduced risk of systemic toxicity.

## 4. Conclusions

In this study, we report that encapsulation of BA in LA-modified-CL-NPs resulted in four-fold increase in the local concentration of BA in the liver relative to its concentration in the blood. In vivo biodistribution studies revealed the superiority of BA-loaded-LA-modified-CL-NPs to target the liver compared to BA-loaded-LA-free-CL-NPs. This study highlights the great potential of using LA-modified-CL-NPs to increase the local bioavailability of therapeutic agents in the liver upon oral administration. Also, This is the first study to highlight the synergistic liver-targeting potential of LA, CL and nanotechnology when used simultaneously in the formation of ultrahigh drug-loaded NPs for targeting the liver. Besides, the proposed LA-modified-CL-NPs formulation offers a mean for ultrahigh encapsulation of therapeutic molecules suffering from poor or pH-dependent water solubility, which are usually difficult to encapsulate using conventional techniques.

## Figures and Tables

**Figure 1 pharmaceutics-12-00107-f001:**
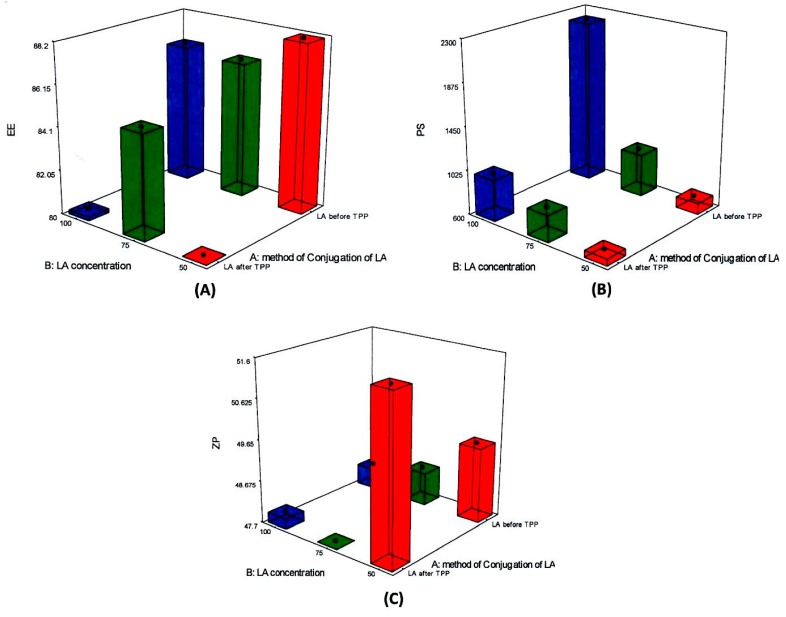
Response plots for the main effects of the method of addition of lactobionic acid (LA)-ethyldimethylcarbodiimide (EDC) to chitosan lactate (CL) solution and the concentration of LA used on (**A**) entrapment efficiency (EE%); (**B**) particle size (PS) and (**C**) Zeta potential (ZP).

**Figure 2 pharmaceutics-12-00107-f002:**
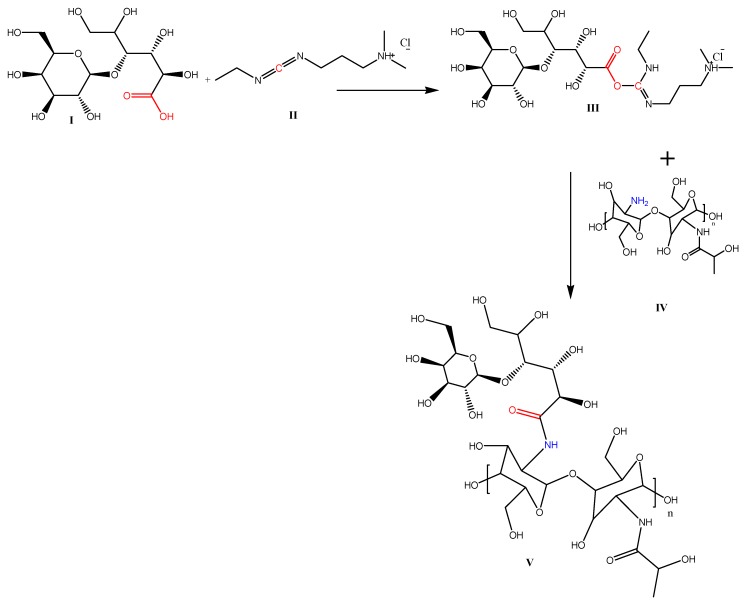
The proposed interaction mechanism between LA (I), EDC (II) and CL (IV) during the formation of LA-modified-CL (V).

**Figure 3 pharmaceutics-12-00107-f003:**
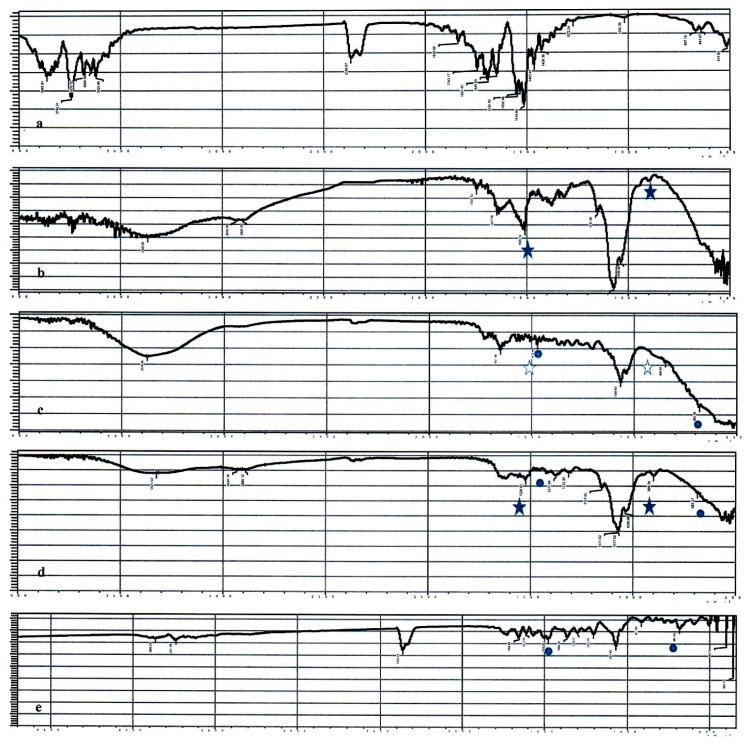
ATR spectra of (**a**) LA; (**b**) CL; (**c**) BA-loaded-LA-modified-CL-NPs; (**d**) BA-loaded-LA-free-CL-NPs and (**e**) BA. The closed stars represent the amine characteristic bands of CL, the open stars represent the absence of the amine characteristic bands of CL and the dots represent the aromatic characteristic bands attributed to the drug.

**Figure 4 pharmaceutics-12-00107-f004:**
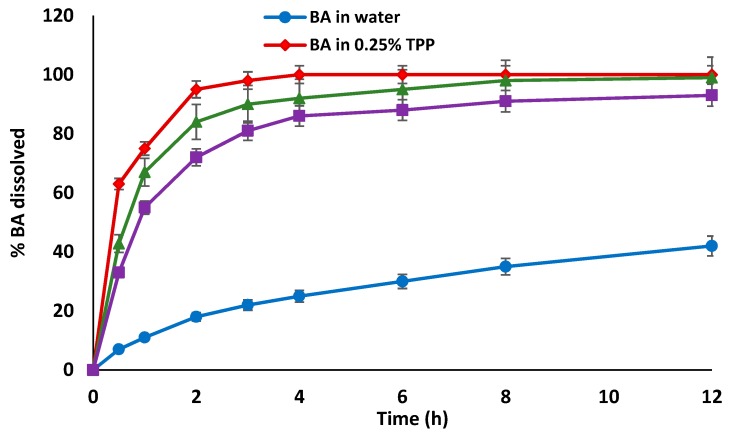
In vitro release profiles of BA from its solution in 0.25% TPP, BA-loaded-LA-free-CL-NPs and BA-loaded-LA-modified-CL-NPs in phosphate buffer (pH 6.8) at 37 °C in comparison to the in vitro release of BA from its dispersion in distilled water. Data points are mean ± SD (*n* = 3).

**Figure 5 pharmaceutics-12-00107-f005:**
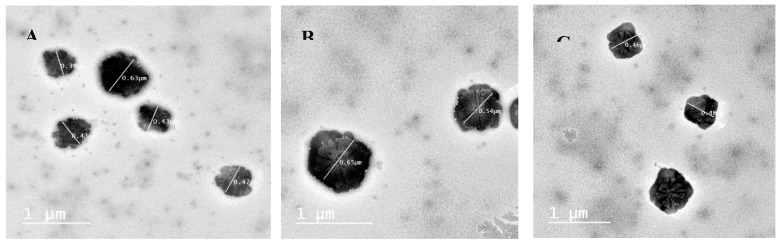
TEM micrographs of (**A**) BA-loaded-LA-free-CL-NPs; (**B**) BA-loaded-LA-modified-CL-NPs and (**C**) BA-loaded-LA-modified-CL-NPs after sonication.

**Figure 6 pharmaceutics-12-00107-f006:**
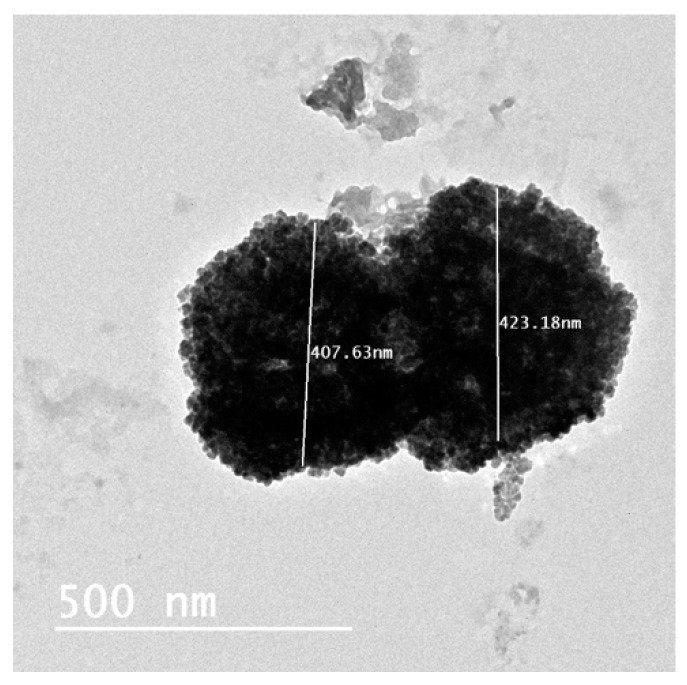
TEM micrograph of BA-loaded-LA-free-CL-NPs after 15-day storage at room temperature showing particles aggregation.

**Figure 7 pharmaceutics-12-00107-f007:**
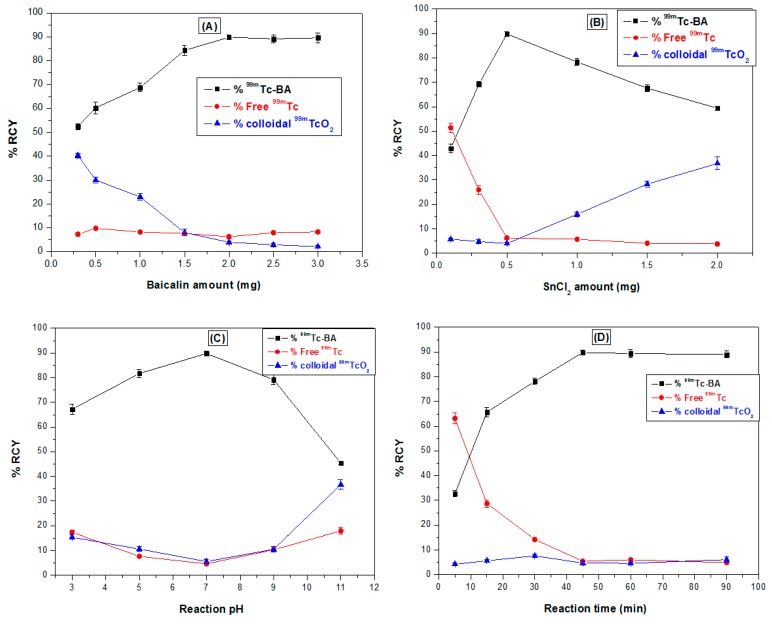
Effect of different labelling conditions on the percentage radiochemical yield (% RCY) of ^99m^Tc-BA complex. (**a**) BA amount, (**b**) SnCl_2_ amount, (**c**) reaction pH and (**d**) reaction time.

**Figure 8 pharmaceutics-12-00107-f008:**
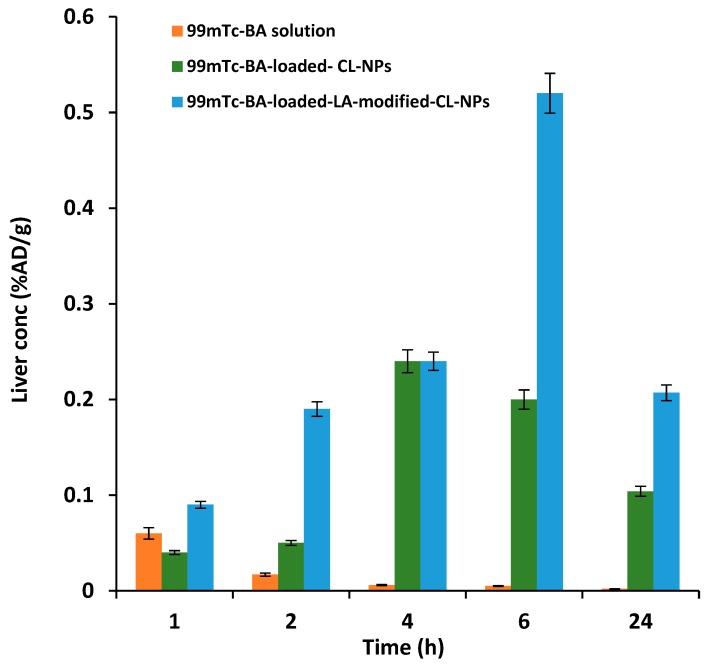
Mean (±SD) BA concentration in liver following oral administration of radiolabeled ^99m^Tc-BA to mice.

**Figure 9 pharmaceutics-12-00107-f009:**
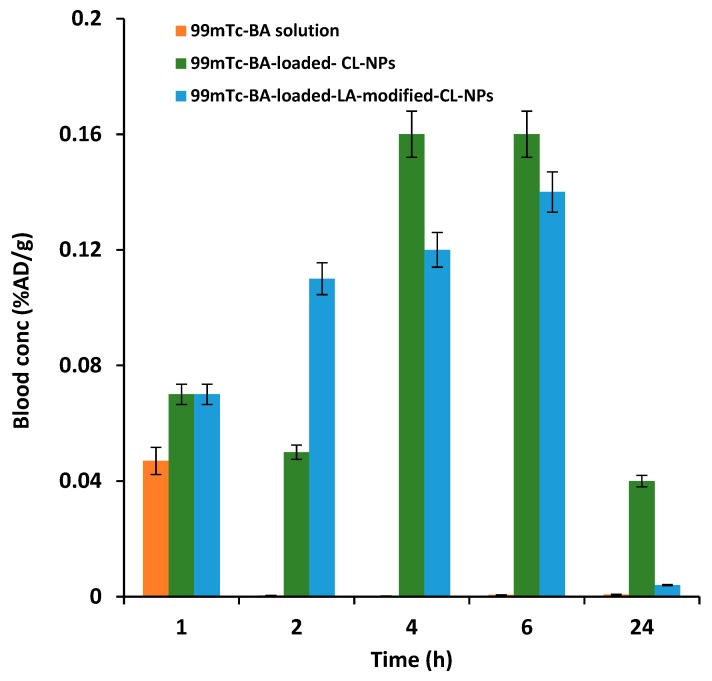
Mean (±SD) BA concentration in the blood following oral administration of radiolabeled ^99m^Tc-BA to mice.

**Table 1 pharmaceutics-12-00107-t001:** The effect of process and formulation variables on the entrapment efficiency (EE%), particle size (PS), zeta potential (ZP) and polydispersity (PDI) of BA-loaded-LA-modified-CL-NPs.

Formulation Code	LA Addition	LA Conc (mg/L)	EE (%)	PS (nm)	ZP (mV)	PDI
F1	After	50	80.7 ± 0.9	713 ± 65	51.0 ± 0.78	0.72 ± 0.0
F2	After	75	84.3 ± 2.5	882 ± 74	47.8 ± 1.98	0.76 ± 0.0
F3	After	100	80.4 ± 0.9	1141 ± 24	47.9 ± 0.21	0.91 ± 0.0
F4	Before	50	93.1 ± 2.4	621 ± 49	48.9 ± 0.04	0.77 ± 0.1
F5	Before	75	85.8 ± 0.2	1160 ± 182	48.7 ± 0.04	0.67 ± 0.0
F6	Before	100	86.2 ± 5.3	2252 ± 394	48.4 ± 1.06	0.51 ± 0.0
BA-loaded-LA-free NPs	-	-	77.4 ± 3.5	485 ± 25	51.6 ± 0.05	0.78 ± 0.0

Data are mean value ± SD (*n* = 3); After: indicates addition of LA after NPs formation; Before: indicates addition of LA before NPs formation.

**Table 2 pharmaceutics-12-00107-t002:** Output results of the factorial design.

Response	R^2^	Adjusted R^2^	Predicted R^2^	Adequate Precision	Significant Terms
EE%	0.938	0.886	80.7±0.98	10.363	X1 (*p* = 0.0001), X1 × X2 (*p* = 0.0343)
PS (nm)	0.976	0.995	84.3±2.54	18.572	X1 (*p* = 0.0004), X2 (*p* < 0.0001), X1 × X2 (*p* = 0.0014)
ZP (mV)	0.893	0.804	77.4±3.52	7.787	X2 (*p* = 0.0016)

**Table 3 pharmaceutics-12-00107-t003:** Effect of sonication on EE%, PS, and ZP of LA-free NPs and LA-modified NPs.

Formulation	EE%	PS (nm)	ZP (mV)	PDI
**BA-loaded-CL-NPs**				
No Sonication	77.4 ± 3.52	485 ± 25	51.6 ± 0.05	0.78 ± 0.02
Sonication	76.6 ± 6.52	435 ± 19	51.4 ± 0.08	0.75 ± 0.02
**BA-loaded-LA-modified-CL-NPs**				
No Sonication	93.2 ± 2.41	621 ± 79	48.9 ± 0.04	0.77 ± 0.12
Sonication	93.7 ± 3.21	490 ± 39	48.1 ± 0.04	0.31 ± 0.23

**Table 4 pharmaceutics-12-00107-t004:** Mean ± (SD) particle size (PS), entrapment efficiency (EE%) and zeta potential (ZP) of BA-loaded-LA-free-CL-NPs and BA-loaded-LA-modified-CL-NPs after 15-day storage at room temperature or 4 °C.

BA-NPs	Storage Conditions	PS (nm)	EE (%)	ZP (mV)
LA-free	Fresh	435 ± 19	76.6 ± 6.5	41.4 ± 0.08
	Room temp.	645 ± 59 ^∗^	78.2 ± 6.6	37.2 ± 0.05 ^∗^
	4 °C	449 ± 15	75.3 ± 3.2	40.3 ± 0.06
LA-modified	Fresh	490 ± 39	93.7 ± 3.2	48.1 ± 0.04
	Room temp.	805 ± 67 ^∗^	94.6 ± 4.3	41.3± 0.04 ^∗^
	4 °C	482 ± 24	92.4 ± 5.1	47.1 ± 0.02

^∗^*P* < 0.05.

**Table 5 pharmaceutics-12-00107-t005:** Hepatic specific delivery parameters of ^99m^Tc-BA-loaded-LA-modified-CL-NPs relative to ^99m^Tc-BA-loaded-LA-free-CL-NPs following oral administration in mice.

Parameter	LA-Modified-NPs	LA-Free-NPs
DTI (1h)	2.25	-
DTI (2h)	3.8	-
DTI (4h)	1	-
DTI (6h)	2.6	-
DTI (24h)	2	-
DTE	4.22	1.45
RTE	2.24	-

**Table 6 pharmaceutics-12-00107-t006:** Mean (±SD) pharmacokinetic parameters of BA in blood and liver tissue following oral administration of radiolabeled ^99m^Tc-BA to mice.

**Blood**	**C_max_ (%AD/g)**	**AUC_0–24_ (%AD/g)**
^99m^Tc-BA solution	0.04 ± 0.006	0.1 ± 0.001
^99m^Tc-BA-loaded-LA-free-CL-NPs	0.16 ± 0.011 ^*,a^	2.43 ± 0.34 ^*,a^
^99m^Tc-BA-loaded-LA-modifie-CL-NPs	0.14 ± 0.014 ^*,a^	1.87 ± 0.025 ^*,a,b^
**Liver**	**C_max_ (%AD/g)**	**AUC_0-24_ (%AD/g)**
^99m^Tc-BA solution	0.07 ± 0.01	0.30 ± 0.04
^99m^Tc-BA-loaded-LA-free-CL-NPs	0.24 ± 0.02 ^*,a^	3.53 ± 0.51 ^*,a^
^99m^Tc-BA-loaded-LA-modified-CL-NPs	0.52 ± 0.06 ^*,a,b^	7.9 ± 0.94 ^*,a,b^

^*^*P* < 0.05; ^a^ Versus BA solution; ^b^ Versus BA-loaded-LA-free-CL-NPs.

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
