# Peer review of "Nanoparticle-Mediated Dual Targeting: An Approach for Enhanced Baicalin Delivery to the Liver"

_pharmaceutics, 2020, doi:10.3390/pharmaceutics12020107_

Round 1

Reviewer 1 Report

In present MS, lactobionic acid (LA) modified chitosan lactate (CL) nanoparticle loaded baicalin (BA) for liver-targeting was prepared and evaluated.

The strategy of targeting asialoglycoprotein (ASGP) receptors hepatic drug delivery is not novel. In addition, for liver-targeting, liver cells should be clear and definite being many kinds of liver cells in liver. More important, the efficiency of this nanoparticles should be evaluated.

Reviewer 2 Report

Ahmed et al report the development and characterization of 99mTc labeled polymeric nanoparticles based on chitosan modified by lactobionic acid for the targeting and drug delivery of baicalin to hepatocytes. The manuscript is well written and should be of interest for the audience of Pharmaceutics.

The reviewer has some comments:

- the authors should check the good homogeneity of the police used in all the manuscript (for example see line 46-47 which is different from line 45 and 48),

- the authors should check typological errors, see for example medicated instead of mediated (line 60), glactose instead of galactose (line 70),

- in the introduction the authors should also add these 2 articles: Richard et al, Int J Pharm 2009, 379, 301-308 ; Chaumet-Riffaud et al, Bioconjugate Chem. 2010, 21, 589-596,

- table 3: how the authors explain the low difference in term of zeta potential for BA-loaded-CL-NPs and BA-loaded-LA-modified-CL-NPs? Don’t we expect to have a higher decrease of the positive charge after modification with LA?

- on figure 2, molecule IV (CL) should be drawn on the arrow under molecule III,

- on figure 3, for clarity, the authors should try to superpose the different spectrum on the same graph,

- did the authors evaluated the therapeutic effect of baicalin either in vitro or in vivo?

Round 2

Reviewer 1 Report

Accept